# Acceptability and feasibility of two interventions in the MooDFOOD Trial: a food-related depression prevention randomised controlled trial in overweight adults with subsyndromal symptoms of depression

Matthew Owens ![ORCID] ,[1] Edward Watkins,[1] Mariska Bot,[2] Ingeborg Annemarie Brouwer,[3] Miquel Roca,[4] Elisabeth Kohls,[5] Brenda Penninx,[2] Gerard van Grootheest,[2] Mieke Cabout,[3] Ulrich Hegerl,[6] Margalida Gili,[4] Marjolein Visser,[3] on behalf of the MooDFOOD Prevention Trial Investigators

For numbered affiliations see end of article.

**Correspondence to**
Dr Matthew Owens;
m.owens-solari@exeter.ac.uk

## ABSTRACT

**Objectives** We report on the acceptability, feasibility, dose-response relationship and adherence of two nutritional strategies to improve mood (multinutrient supplements; food-related behavioural activation (F-BA)) studied in a randomised controlled depression prevention trial (the Multi-country cOllaborative project on the rOle of Diet, Food-related behaviour, and Obesity in the prevention of Depression (MooDFOOD) Trial). We also assessed baseline determinants of adherence and assessed whether better adherence resulted in lower depressive symptoms.

**Design** Randomised controlled trial with a 2×2 factorial design conducted between 2015 and 2017.

**Setting** Germany, the Netherlands, UK and Spain.

**Participants** Community sample of 1025 overweight adults with elevated depressive symptoms without a current episode of major depressive disorder. Main eligibility criteria included age (18–75 years), being overweight or obese, and having at least mild depressive symptoms, shown by a Patient Health Questionnaire Score of ≥5. A total of 76% of the sample was retained at the 12-month follow-up.

**Interventions** Daily nutritional supplements versus pill placebo or an F-BA therapy, delivered in individual and group sessions versus no behavioural intervention over a 1-year period.

**Primary and secondary outcome measures** Primary outcome: self-reported acceptability of the interventions. Secondary outcomes: adherence and self-reported depressive symptoms.

**Results** Most participants reported that the F-BA was acceptable (83.61%), feasible to do (65.91%) and would recommend it to a friend (84.57%). Individual F-BA sessions (88.10%) were significantly more often rated as positive than group F-BA sessions (70.17%) and supplements (28.59%). There were statistically significant reductions in depressive symptoms for those who both adhered to the F-BA intervention and had a history of depression (B=−0.08, SE=0.03, p=0.012) versus those

## Strengths and limitations of this study

► A large sample from a multicountry European prospective randomised controlled depression prevention trial was used.

► There have thus far been very few studies assessing the acceptability and feasibility of nutritionally based depression prevention interventions.

► The representativeness of the sample (overweight, mild depressive symptoms) may limit the generalisability of the conclusions.

► The mediating mechanisms underlying potential reductions in depressive symptoms are as yet untested, making it difficult to identify key targets for future interventions.

who had no history of depression. Supplement intake had no effect on depressive symptoms irrespective of adherence.

**Conclusions** F-BA may have scope for development as a depression prevention intervention and public health strategy but further refinement and testing are needed.

**Trial registration number** NCT02529423.

## INTRODUCTION

Depression is currently the leading global health problem[1] with associated annual costs estimated at $1 trillion.[2] Similarly, obesity has become a global pandemic,[3] which as well as being associated with a range of physical health conditions such as diabetes,[4] may also increase the risk for depression.[5] Given the high prevalence of major depressive episodes (MDE) and major depressive disorder (MDD),[6 7] depression prevention interventions have now become a global priority.[8 9]

One emerging and promising approach to depression prevention falls under the 'nutritional psychology' framework, which suggests that change in diet, nutrition and food-related behaviour may prevent depression.[10 11] Recent meta-analyses have provided robust observational evidence of the inverse association between diet and depression, showing that higher adherence to a Mediterranean diet is linked to a lower risk of depression.[12 13] The data from these meta-analyses included cross-sectional and longitudinal studies: prospective and in particular randomised controlled prevention trials are necessary to test the causal direction of the relationship between diet and depression. Several randomised controlled trials (RCTs) have been conducted demonstrating a positive effect of dietary intervention strategies either in individuals with major depression or with elevated symptoms of depression,[14–16] however, there is still a limited number of trials examining the effect of nutritional strategies on the prevention of depression.

One recent attempt to conduct a large scale RCT of diet and nutrition to prevent depression is the Multi-country cOllaborative project on the rOle of Diet, Food-related behaviour, and Obesity in the prevention of Depression (MooDFOOD) Trial.[17] Participants in this 2×2 factorial designed trial were randomised to daily nutritional supplementation versus pill placebo, and to a nutritionally based psychological therapy; a multimodal food-related behavioural activation (F-BA) intervention versus no behavioural intervention. The F-BA mode of delivery was comprised of a mix of individual and group sessions, designed such that all participants started with individual sessions and then progressed to group sessions for maintenance of the intervention.[17]

The MooDFOOD Trial found that while neither intervention had significant effects on incidence of MDE over 12 months,[17] F-BA, relative to no F-BA, reduced symptoms of depression at 12 months follow-up for those with higher initial severity and prevented MDE for those with higher levels of treatment adherence,[17] as well as reducing unhealthy food consumption and increasing healthy food consumption.[18] There were no beneficial effects of daily nutrient supplementation on depressive symptoms, and there was even evidence that outcomes were worse for those taking supplements versus placebo.

However, examining intervention outcomes alone does not provide relevant information to evaluate the real world implementation and dissemination of an intervention, such as whether and in what form patients will use the intervention, how much patients find the intervention helpful and engaging, how adherent patients are with the intervention (ie, number of treatment sessions attended or percentage of pills taken) and whether the dose of therapy received influences its outcomes. These are important questions to determine whether the therapy can be implemented beyond the trial, whether it would be used and what influences its effects. The Medical Research Council guidance on delivering complex interventions makes it clear that assessing the feasibility of delivering a complex intervention is an essential but often overlooked step and that interventions are hampered by a range of problems including those around acceptability, adherence to intervention and delivery.[19]

Assessing whether community samples and those at risk of common mental health problems such as depression are likely to adhere to nutrition-based interventions to reduce symptoms or increase well-being is an important next step, and while work on this issue has begun,[14–17] more in-depth analysis is needed.

To address this critical issue, we report on the two interventions in more depth, in particular exploring their adherence and acceptability as well as testing for any dose-response relationships and assessing potential for adherence (both unmoderated and moderated by baseline characteristics) to influence depressive symptoms over time. We note that in the main trial outcome analysis[17] there was no evidence to suggest that the supplements had beneficial effects and that they may cause harm. With this in mind, we concentrated on the following hypotheses:

1. Acceptability. Do participants value and like the preventive nutritional intervention? This is critical, as without acceptability, the intervention is unlikely to be taken up and used by the target population in the future. Because the trial was advertised openly as an investigation into two interventions with equal potential to help with depression, we hypothesised that both would be equally acceptable to participants. We hypothesised that individual sessions would be favoured more than group sessions because of their individualised and private nature and because individual sessions come earlier in the F-BA treatment.

2. Adherence. For the F-BA, we focused on attending F-BA sessions and adherence to the designed elements of the intervention in the current analysis. That is, to what extent were participants doing what the intervention intended? Here we describe the number of sessions attended and the elements of F-BA that were tried and reported as successful by participants. Adherence to the pills was defined as being present if 70% of pills were taken over the course of the trial.

3. Dose effects of adherence. Is there a relationship between the number of F-BA sessions attended and change in depressive symptoms? This is important to know in terms of planning the minimum dose for future implementation. We examined two hypotheses: (1) A higher total number of individual and group sessions attended would predict more reduction of depression symptoms from beginning to end of trial. (2) Only those participants reaching our trial prespecified level of F-BA sessions required for adherence (8/21 sessions, see Method section) would see a significant average reduction in depressive symptoms. We focused primarily on dose effects of F-BA as previously there were no significant beneficial effects on depressive symptoms found for supplements,[17] and report the effects for supplements here as a comparison analysis.

4. Are there baseline determinants of higher adherence and are there significant moderators of the effect of adherence on outcome? That is, are there different populations or subgroups that may be most likely to benefit from the intervention? Given previous findings implicating either 'main effects' of the MooDFOOD intervention on dietary behaviour (eg, Grasso *et al*[18]) or interaction effects between intervention, adherence and pre-existing depression and symptom levels[17] we hypothesised a priori that adherence to intervention has a stronger effect on reducing depressive symptoms for those individuals with (1) A history of clinical depression. (2) Higher baseline depressive symptoms. (3) Lower baseline diet quality scores.

## METHOD
### Participants
The present study used the data from the MooDFOOD randomised controlled prevention trial[17] cohort of adults (n=1025, mean age=46.51 years, range=18–75; mean Patient Health Questionnaire (PHQ-9) Score 7.42; mean body mass index (BMI)=31.38) recruited from four sites in European countries (the Netherlands (NL), Spain (ES), Germany (DE) and the UK). The trial followed a 2×2 factorial design testing the effect of multinutrient supplements and/or food-related behavioural change on the prevention of depression and psychological well-being over a period of 1 year. Main eligibility criteria included being aged between 18 years and 75 years, being overweight or obese (BMI between $25\,kg/m^2$ and $40\,kg/m^2$) and having at least mild depressive symptoms as operationalised by a PHQ Score of ≥5. Participants in the trial were allocated to one of four arms: F-BA plus supplements (n=256), no F-BA plus supplements (n=256), F-BA plus placebo (n=256), no F-BA plus supplements (n=256) or no F-BA plus placebo (n=257).

The trial was conducted according to Helsinki good practice guidelines.[20]

### Patient and public involvement
The study was supported by a volunteer public advisory group in the UK which provided input to the programme of research at the planning stage. This volunteer group primarily advised us on the design of the informational material to support the trial and the consent to participate process and forms. A patient/lay person representative (Germany) served as a member of our Trial Steering Committee for the duration of the trial.

### F-BA therapy
The F-BA consisted of a protocol-based intervention incorporating standard approaches of BA (behavioural activation) proven effective in depression treatment[21] including self-monitoring, functional analysis and activity scheduling, with a focus on changing dietary habits to improve mood. F-BA was provided by trained psychologists familiar with BA (15 individual sessions, 6 group sessions, for a 1-year period) and a dietician was

available to all study sites to provide advice. The intervention was delivered over a maximum of 21 sessions with up to 15 half-hour individual sessions, provided in single or double (1 hour) meetings. Individual sessions were initially given weekly (sessions 1–8) and then held every 2 weeks (sessions 9–15). These were followed by up to six group sessions (maximum of 10 participants per group and lasting approximately 1 hour) occurring monthly initially (sessions 16–18) and finally once every 2 months (sessions 19–21). A range of potential strategies was employed including food-related strategies and food group targets to aim for. These were introduced and explained to participants in the course of the first eight sessions and subsequently revisited and reviewed over the remaining sessions. The F-BA was designed to start with individual sessions, leading on to group sessions, with the former focused on individualised review of food and mood habits and planning of new approaches, and the latter as consolidation and maintenance of what was learnt. This ordering of individual sessions followed by group sessions was fixed. F-BA followed principles of introducing participants to key information and then re-inforcing and consolidating concepts over time.

### Intervention with pills
Patients received either multinutrient supplements (1412 mg of eicosapentaenoic and docosahexaenoic omega-3 fatty acids (ratio 3:1), 30 µg selenium, 400 µg folic acid, and 20 µg vitamin D3 coupled with 100 mg calcium) or placebos provided in two pills per day, to be taken daily for 1 year. Further details on both interventions have been reported elsewhere.[17 22]

### Measures
Measures were collected at baseline (T0), 3 months (T3), 6 months (T6) and 12 months (T12; end of trial). Bespoke trial acceptability questions were completed at the final 12-month assessment by participants who completed the trial. Participants were asked standardised questions about their experiences and opinions on the trial supplements/placebo. For example, '*How satisfied were you with the supplements?*' Similarly, questions relating to the F-BA included '*How helpful was the coaching in improving your mood?*' These questions are itemised in table 1. In addition, participants of the F-BA intervention were asked which mood-related strategies and food targets they had tried and the extent to which they believed they were successful.

### Diet quality
Dietary intake was assessed with the 250-item Global Allergy and Asthma European Network of Excellence (GA2LEN) food frequency questionnaire.[23] Intake of the 11 MooDFOOD diet quality food groups was summed to obtain a MooDFOOD diet quality score (range: 0 indicating poor adherence, 77 indicating optimal adherence).

### Depressive symptoms—PHQ-9
The PHQ-9[24] is a well-validated nine-item self-report instrument that assesses current symptoms of depression

**Table 1** Bespoke self-report questions on acceptability of interventions

| Questionnaire item | Response scale |
| --- | --- |
| **F-BA overall** | |
| To what extent has the coaching met your needs? | 5-point Likert not at all to completely |
| How helpful was the coaching in improving your mood? | 5-point Likert very unhelpful to very helpful |
| To what extent was it worth your time doing the coaching? | 5-point Likert very unworthwhile to very worthwhile |
| Overall, how satisfied were you with the coaching? | 5-point Likert very unsatisfied to very satisfied |
| Overall, how easy was it for you to comply with the coaching? | 5-point Likert very easy to very difficult |
| If a friend was in need of similar help, would you recommend the coaching to a friend? | Binary (yes/no) |
| How satisfied were you with the amount of homework between sessions? | 5-point Likert very unsatisfied to very satisfied |
| How satisfied were you with the frequency of the sessions? | 5-point Likert very unsatisfied to very satisfied |
| How satisfied were you with the length of the sessions? | 5-point Likert very unsatisfied to very satisfied |
| **F-BA individual sessions** | |
| How satisfied were you with the individual sessions? | 5-point Likert very unsatisfied to very satisfied |
| How helpful were the individual sessions in improving your mood? | 5-point Likert very unhelpful to very helpful |
| How helpful were the individual sessions in increasing helpful habits? | 5-point Likert very unhelpful to very helpful |
| How helpful were the individual sessions in reducing unhelpful habits? | 5-point Likert very unhelpful to very helpful |
| How helpful were the individual sessions in addressing your goals for F-BA | 5-point Likert very unhelpful to very helpful |
| How satisfied were you with the group sessions? | 5-point Likert very unsatisfied to very satisfied |
| How helpful were the group sessions in improving your mood? | 5-point Likert very unhelpful to very helpful |
| How helpful were the group sessions in addressing your goals for the coaching? | 5-point Likert very unhelpful to very helpful |
| How helpful were the group sessions in increasing helpful habits? | 5-point Likert very unhelpful to very helpful |
| How helpful were the group sessions in reducing unhelpful habits? | 5-point Likert very unhelpful to very helpful |
| **Supplements** | |
| How helpful were the supplements in improving your mood? | 5-point Likert very unhelpful to very helpful |
| Overall, how satisfied were you with the supplements? | 5-point Likert very unsatisfied to very satisfied |
| How easy was it to comply with the regimen for the capsules from the jar with the blue label? | 5-point Likert very hard to very easy |
| How easy was it to comply with the regimen for the capsules from the jar with the green label? | 5-point Likert very hard to very easy |
| If a friend was in need of similar help, would you recommend the supplements to them? | Binary (yes/no) |

F-BA, food-related behavioural activation.

over the last 2 weeks based on Diagnostic and Statistical Manual of Mental Disorders - Fourth Edition criteria for major depression, which is used extensively in primary care and clinical trials and is proven to be as good at detecting depression as clinician-administered instruments.[25]

## Major depressive disorder

At baseline, a previous history of MDD was measured using a brief standardised diagnostic interview (duration of 15–30 min): the Major Depression module from the Mini International Neuropsychiatric Interview.[26]

## Measuring adherence to intervention

Adherence to the F-BA intervention was defined as having attended at least 8 out of the 21 sessions. The adherence criterion for the pills was 70% of pills taken over the course of the trial. The latter calculation was based on weights of provided versus returned supplement containers as well as self-reported supplement use.[17]

We also assessed the proportion of F-BA participants who reported attempting either mood-related strategies or food-related strategies and food targets as promoted in the F-BA intervention, as well as the degree to which these attempts were successful. Better adherence to the F-BA

would be evidenced by higher proportions of participants attempting strategies and reporting them as successful.

## Statistical approach

1. Acceptability. We calculated the percentage of self-report responses to acceptability questions for both pills and F-BA psychological therapy. We compared differences between groups (supplement vs F-BA, individual vs group sessions) using $\chi^2$ tests of association.
2. Adherence. We reported the levels of treatment adherence for pills and F-BA and explored differences in influences in adherence by baseline participant characteristics (gender, age, past history of depression, depressive symptoms, diet quality and BMI) by adherence using $\chi^2$ and t-tests.
3. Dose effects of adherence. We used the total number of F-BA sessions, individual and group separately, as well as the binary variable for overall good F-BA adherence, to predict the longitudinal change in depressive symptoms from baseline (T0) to trial follow-up at 12 months (T12). The dose effects of adherence for pills were assessed using the binary pill adherence variable. These relationships were tested using linear, quadratic and cubic polynomials in linear regression models. T12 depressive symptom level was the outcome, adjusted for T0 depressive symptoms, history of depression, sex and trial site.
4. Moderators of the effect of adherence on outcome. We tested interactions between good F-BA adherence/pill adherence and the three potential moderators outlined in hypothesis 4 (history of depression, baseline depressive symptoms and diet quality score). This was achieved by using separate linear regression models with T12 depressive symptoms as the outcome adjusted for T0 depressive symptoms and including the adherence variables (binary), the three potential moderators and the interaction terms between adherence and the potential moderators. Only significant interactions were followed up. We decomposed any significant interactions by assessing the influence of adherence at different levels of a significant moderator. For the binary moderator (a history of depression) we tested the effect of adherence on outcome for those with a history of depression versus those without. For continuous variables (baseline depressive symptoms and diet quality) we planned to test the effect of adherence at levels of the moderator, equal to and above the median versus below the median average. We also specified a model that allowed us to test an alternative hypothesis that change in depressive symptoms would lead to poor adherence to intervention over the trial. In this model, change in depressive symptoms was specified from T0 to the earliest possible time point (T3) to maximise the temporal precedence of depressive symptoms over adherence.

The models were run in the Mplus programme using full information maximum likelihood estimation to account for missing data, which is a robust unbiased and efficient technique outperforming traditional approaches to missing data.[27 28] Consistent with views expressed elsewhere,[29] our approach here is to simply report, describe and discuss any tests of significance carried out in order to maintain a balance between type I and type II error rates, rather than apply any corrections to p values. We encourage the reader to evaluate the size of effect and regard all findings as tentative until further corroboration.[30]

## RESULTS

### Acceptability of the MooDFOOD interventions

#### Food-related behavioural activation

A total of 76% of the sample was retained at the 12-month follow-up. The F-BA acceptability questionnaire data were available for a maximum of 60.7% of the F-BA group (311 from a possible 512 individuals randomised). The remainder of participants had either dropped out of the trial and were not available to complete an end of trial questionnaire pack or choose not to answer questions on the F-BA. The results showed that the F-BA was well received by participants, with 68.2% saying F-BA had either met their needs completely or a lot, 18.7% said somewhat and 13.2% said slightly or not at all. Most participants (83.6%) said they were either very satisfied or somewhat satisfied with the F-BA intervention, overall. The majority reported that 'it was easy to do' (65.9%), 'worth their time doing it' (84.9%) and 'most would recommend the F-BA to a friend in similar need' (84.6%). The majority of participants was also satisfied with both the length (82.3%) and frequency (68.5%) of F-BA sessions. Most participants also reported that the burden of homework was 'just right' (67.9%) with 27.7% preferring less and 4.5% preferring more. Individual sessions were rated significantly more positively than group sessions in terms of overall participant satisfaction ($\chi^2$=67.70, p<0.001), help with improving mood ($\chi^2$=39.96, p<0.001), increasing helpful habits ($\chi^2$=54.41, p<0.001) and decreasing unhelpful habits ($\chi^2$=49.97, p<0.001) and being helpful in addressing participants' goals ($\chi^2$=39.13, p<0.001). See table 2 for details.

#### F-BA strategies, behaviours and targets

The majority of behaviour strategies had been attempted by most F-BA participants (70%–86% had tried each strategy). The top three most tried strategies were *limit your snacks* (86.8%), *pay attention to habitual snacking* (84.6%) and *avoid emotional eating* (81.7%). See table 3 for details. Most participants reported that attempting the food-related strategies and targets was to some extent (*somewhat* or *very*) successful (91.2%–99.6%). The most successful food-related strategy tried (*very successful*) included *planning shopping trips* (67.6%), *eating three regular main meals per day* (66.7%) and *controlling impulses when shopping* (55.5%).

Similarly, most participants had tried to increase or limit the consumption of certain food groups (54%–82%).

**Table 2** Self-report rating of individual and group F-BA sessions and supplement/placebo use

| Questionnaire item | Freq (%) | Freq (%) | Freq (%) | Freq (%) | Freq (%) | $\chi^2$ |
|---|---|---|---|---|---|---|
| **Overall, how satisfied were you with the intervention?** | **Very unsatisfied** | **Somewhat unsatisfied** | **Neutral** | **Somewhat satisfied** | **Very satisfied** | |
| F-BA individual (n=311) | 2 (0.64) | 10 (3.22) | 25 (8.04) | 93 (29.90) | 181 (58.20) | |
| F-BA group (n=228) | 3 (1.32) | 31 (13.60) | 34 (14.91) | 104 (45.61) | 56 (24.56) | F-BA $\chi^2$=67.70 (4)** |
| Supplements (n=759) | 26 (3.43) | 43 (5.67) | 473 (62.32) | 147 (19.37) | 70 (9.22) | Tx $\chi^2$=385.36 (4)** |
| Pills (n=371) | 15 (4.04) | 17 (4.58) | 216 (58.22) | 87 (23.45) | 36 (9.70) | |
| Placebo (n=388) | 11 (2.84) | 26 (6.70) | 257 (66.24) | 60 (15.46) | 34 (8.76) | Pills $\chi^2$=10.69 (4)* |
| **How helpful were the interventions in improving your mood?** | **Very unhelpful** | **Somewhat unhelpful** | **Neutral** | **Somewhat helpful** | **Very helpful** | |
| F-BA individual (n=311) | 3 (0.96) | 6 (1.93) | 54 (17.36) | 123 (39.55) | 125 (40.19) | |
| F-BA group (n=228) | 7 (3.07) | 18 (7.89) | 66 (28.95) | 93 (40.79) | 44 (19.30) | F-BA $\chi^2$=39.96 (4)** |
| Supplements (n=759) | 51 (6.72) | 22 (2.90) | 512 (67.46) | 131 (17.26) | 43 (5.67) | Tx $\chi^2$=333.60 (4)** |
| Pills (n=371) | 16 (4.31) | 9 (2.43) | 247 (66.58) | 74 (19.95) | 25 (6.74) | |
| Placebo (n=388) | 35 (9.02) | 13 (3.35) | 265 (68.30) | 57 (14.69) | 18 (4.64) | Pills $\chi^2$=11.41 (4)* |
| **How helpful were the sessions in increasing helpful habits?** | **Very unhelpful** | **Somewhat unhelpful** | **Neutral** | **Somewhat helpful** | **Very helpful** | |
| F-BA individual (n=311) | 1 (0.32) | 5 (1.61) | 33 (10.61) | 141 (45.34) | 131 (42.12) | |
| F-BA group (n=228) | 6 (2.63) | 16 (7.02) | 60 (26.32) | 101 (44.30) | 45 (19.74) | F-BA $\chi^2$=54.31 (4)** |
| **How helpful were the sessions in decreasing unhelpful habits?** | **Very unhelpful** | **Very unhelpful** | **Neutral** | **Somewhat helpful** | **Very helpful** | |
| F-BA individual (n=311) | 3 (0.96) | 6 (1.93) | 34 (10.93) | 173 (55.63) | 95 (30.55) | |
| F-BA group (n=228) | 6 (2.63) | 14 (6.14) | 70 (30.70) | 102 (44.74) | 36 (15.79) | F-BA $\chi^2$=49.97 (4)** |
| **Were the sessions helpful in addressing your goals?** | **Very unhelpful** | **Very unhelpful** | **Neutral** | **Somewhat helpful** | **Very helpful** | |
| F-BA individual (n=311) | 3 (0.96) | 9 (2.89) | 35 (11.25) | 140 (45.02) | 124 (39.87) | |
| F-BA group(n=228) | 6 (2.63) | 20 (8.77) | 54 (23.68) | 102 (44.74) | 46 (20.18) | F-BA $\chi^2$=39.13 (4)** |

F-BA $\chi^2$ statistics refer to the difference between individual and group F-BA overall ratings. Tx $\chi^2$ statistics refer to the difference between individual F-BA and the supplement group overall ratings. Pills $\chi^2$ statistics refer to the difference between pill and placebo groups.
Results were unchanged when using Fisher's exact test for small cell sizes.
*p<0.05, **p<0.001.
F-BA, food-related behavioural activation.

The most tried food group targets were *eating 300–400 g of vegetables* (82.6%), *eating two to three pieces of fruit per day* (81.0%) and *limiting intake of processed food and drinks* (81.0%). For food group targets, those reported as being most successful (*very successful*) included *using olive oil as the principle fat source for cooking* (78.6%), *drinking alcoholic beverages in moderation* (72.3%) and *choosing wholegrain products* (65.2%) (see table 3).

### Multinutrient supplement and placebo pills

Participants reported that it was easy to comply with the daily pill intervention (supplement pill or placebo) (75.4%), although in contrast to the F-BA results, most said that they would not recommend them to a friend in similar need (55.6% No, 44.4% Yes). Participants most often reported a 'neutral' response to how satisfied they were with the supplements (62.3%), with 28.6% to some extent satisfied and 9.1% to some extent dissatisfied. Similarly, the majority was 'neutral' when asked how helpful

the supplements were in improving mood (67.5%), where 22.9% said they were to some extent helpful and 9.6% said they were to some extent unhelpful. Participants were significantly less positive about the supplements than the individual F-BA sessions on all equivalent questions including on whether they would refer a friend to the intervention (ie, 55.6% vs 84.6% Yes, $\chi^2$=146.40, df (1), p<0.001), their overall satisfaction with the intervention ($\chi^2$=385.36, p<0.001) and how helpful the intervention was in improving mood ($\chi^2$=333.60, p<0.001).

### F-BA adherence

A total of 365 participants (71.3%) allocated to F-BA met criteria for good F-BA adherence (a priori criterion of a minimum of 8 out of 21 F-BA sessions attended). The individual sessions had an attendance rate of 70.4% and the group sessions had an attendance of 28.8%; 46.7% of F-BA participants attended all 15 individual sessions, while 53.3% did not attend any group sessions. No

**Table 3** F-BA strategies and targets tried and participant-reported success (n=168 to 270)

| Food-related strategy | N (%) tried strategy | Very unsuccessful N (%) | Somewhat successful N (%) | Very successful N (%) |
|---|---|---|---|---|
| Eat three regular main meals per day | 246 (79.10) | 8 (3.25) | 74 (30.08) | 164 (66.67) |
| Pay attention to food when eating | 246 (79.10) | 12 (4.88) | 128 (52.03) | 106 (43.09) |
| Limit your snacks | 270 (86.82) | 14 (5.19) | 139 (51.48) | 117 (43.33) |
| Pay attention to habitual snacking | 263 (84.57) | 13 (4.94) | 126 (47.91) | 124 (47.15) |
| Avoid emotional eating | 254 (81.67) | 14 (5.51) | 120 (47.24) | 120 (47.24) |
| Plan your shopping trips | 219 (70.42) | 10 (4.57) | 61 (27.85) | 148 (67.58) |
| Control your impulses when shopping | 229 (73.63) | 7 (3.06) | 95 (41.48) | 127 (55.46) |
| Explore and expand your cooking skills | 227 (72.99) | 9 (3.96) | 94 (41.41) | 124 (54.63) |
| **Food- targets** | | | | |
| Eat 300–400 g of vegetables | 257 (82.64) | 9 (3.50) | 127 (49.42) | 121 (47.08) |
| Eat 2–3 pieces of fruit per day | 252 (81.03) | 10 (3.97) | 94 (37.30) | 108 (58.73) |
| Eat three times fish per week | 194 (62.38) | 16 (8.25) | 118 (60.82) | 60 (30.93) |
| Reduce your meat intake to 300 g per week | 209 (67.20) | 6 (2.87) | 85 (40.67) | 118 (56.46) |
| Eat pulses or legumes three times per week | 170 (54.66) | 15 (8.82) | 84 (49.41) | 71 (41.76) |
| Choose wholegrain products | 244 (78.46) | 5 (2.05) | 80 (32.79) | 159 (65.16) |
| Use three servings of low-fat dairy products per day | 168 (54.02) | 6 (3.57) | 75 (44.64) | 87 (51.79) |
| Use olive oil as your principle fat source for cooking | 234 (75.24) | 1 (0.43) | 49 (20.94) | 184 (78.63) |
| Limit your intake of processed foods and soft drinks | 252 (81.03) | 6 (2.38) | 86 (34.13) | 160 (63.49) |
| Drink alcoholic beverages in moderation | 213 (68.49) | 4 (1.88) | 55 (25.82) | 154 (72.30) |

F-BA, food-related behavioural activation.

non-compliers attended a group session but 65.5% of compliers attended one or more group sessions. The number of sessions attended was as follows: individual session mean=10.56, SD=5.49, range=0–15; group session mean=1.73, SD=2.15, range=0–6.

To illustrate the relationship between individual and group session adherence, we created two binary variables representing 'high' and 'low' adherence for individual and group sessions, respectively. In the 'high' individual session category, 281 participants (54.88%) attended either 14 or 15 (the maximum) sessions. In the 'low' individual session category, 231 participants (45.12%) attended 0–13 sessions (15.58% attended 0 sessions). In the high group session category 177 (34.57%) attended 3–6 sessions, whereas in the low group, 335 (65.43%) attended 0–2 sessions (81.49% attended 0 sessions). Of those classified as 'low' individual session adherers, the vast majority (220, 95.24%) was also classified as 'low' group session adherers versus 'high' group adherers (11, 4.76%). Of those classified as 'high' individual session adherers, 115 (40.93%) were classified as 'low' and 166 (59.07%) as 'high' group session adherers.

### Baseline differences by adherence groups
There was no evidence that good F-BA adherence (8/21 sessions) differed by prebaseline characteristics. The proportions of adherence were similar for men and women (men=70.8%, women=71.4%; $\chi^2$=0.02, df (1), p=0.90) and for those with and without a history of depression (MDD=71.5%, no MDD=71.2%; $\chi^2$=0.01, df (1), p=0.94). Good F-BA adherence versus non-adherence did not differ by baseline levels of depressive symptoms (PHQ Scores for not adherent=7.01, SD=4.33, PHQ Scores for adherent=7.29, SD=4.19; t(501)=−0.66, p=0.51), age (not adherent=46.23, SD=12.85, adherent=46.75, SD=12.73; t(510)=−0.41, p=0.68), BMI (not adherent=31.51, SD=3.97, adherent=31.40, SD=3.89; t(510)=0.28, p=0.78) or diet quality (not adherent=50.86, SD=6.58, adherent=51.94, SD=6.98; t(476)=−1.49, p=0.14).

### Dose-response effect of F-BA therapy
There was no direct relationship between the number of F-BA sessions and change in depressive symptoms for individual (linear=B=−0.20, SE=0.57, p=0.73; quadratic=B=0.01, SE=0.08, p=0.93; cubic=B=0.00, SE=0.34, p=0.99) or group sessions (linear=B=−0.33, SE=0.76, p=0.66; quadratic=B=0.07, SE=0.34, p=0.83; cubic=B=−0.55, SE=3.79, p=0.89). Similarly, there was no relationship between good F-BA adherence (binary) and change in depressive symptoms (B=−0.48, SE=0.49, p=0.33).

## Baseline moderators of the effect of adherence on outcome

There was a significant interaction between good F-BA adherence and a history of depression (B=−3.12, SE=1.07, p=0.003) such that adhering to F-BA led to a reduction in depressive symptoms only for those with a history of depression (B=−1.29, SE=0.51, p=0.012). There were no significant interactions between good F-BA adherence and baseline levels of depressive symptoms (B=0.13, SE=0.12, p=0.26), baseline MooDFOOD diet quality levels (B=−1.01, SE=0.08, p=0.19) or trial site (UK=B=−0.21, SE=1.18, p=0.86; ES=B=−2.22, SE=3.65, p=0.54; NL=B=2.08, SE=1.71, p=0.09). In post hoc analysis, we also tested an alternative model where change in depressive symptoms from T0 to T3 was allowed to predict overall adherence to F-BA. This model revealed no significant association (B=−0.00, SE=0.02, p=0.87).

Because we found evidence for the hypothesised significant interaction between good F-BA adherence and a history of depression, we explored the linear dose-response relationship between the combined number of F-BA sessions and a reduction of depressive symptoms, in those with a history of depression. There was a significant association between total number of F-BA sessions and depressive symptoms for those with a history of depression, such that for every additional F-BA session attended, there was an associated average reduction in depressive symptoms (B=−0.08, SE=0.03, p=0.012).

## Adherence to supplement and placebo pills

For pill compliance (supplement or placebo), 513 participants (78.8%) were categorised as adherent (defined a priori as taking ≥70% of the supplements during the 12 months).

Men were more adherent to supplement pills than women (men=88.89% adherence, women=75.46% adherent; $\chi^2$=13.14, df (1), p<0.001). There were no differences on adherence for those with and without a history of depression (MDD=80.09%, no MDD=76.17%; $\chi^2$=1.32, df (1), p=0.25). Pill adherence versus non-adherence did not differ by baseline depressive symptoms (not adherent=6.79, SD=3.79, adherent=7.45, SD=4.29; t(643)=−1.63, p=0.10), average age (not adherent=47.40, SD=11.68; adherent=48.08, SD=13.34; t(649)=−0.55, p=0.59), BMI (not adherent=31.31, SD=3.80, adherent=31.12, SD=3.91; t(649)=0.53, p=0.59) or baseline diet quality (not adherent=51.98, SD=6.12; adherent=51.37, SD=7.29; t(634)=0.86, p=0.39). There was no significant interaction effect on the reduction of depressive symptoms between pill adherence and a history of depression (B=−1.99, SE=0.74, p=0.10), baseline depressive symptoms (B=0.13, SE=0.12, p=0.26), baseline MooDFOOD diet levels (B=0.02, SE=0.06, p=0.78) or trial site (UK=B=0.41, SE=1.08, p=0.70; ES=B=0.21, SE=1.17, p=0.86; NL=B=2.08, SE=1.31, p=0.23). There was no dose-response effect of pills on depressive symptoms (B=−0.55, SE=0.37, p=0.14).

## DISCUSSION

The results suggest that while both MooDFOOD interventions (F-BA and daily nutritional supplementation) were reasonably acceptable to study participants, they reported being significantly more satisfied with the F-BA than the supplements. Participants reported that F-BA was significantly more helpful in improving mood and were significantly more likely to refer a friend in similar need. A large majority of those reporting on F-BA activities said that they had attempted the mood and food-related strategies contained within it and that these attempts had been to some extent successful. These findings are ostensibly to some extent contrary to our findings published elsewhere on the effect of the F-BA on food intake using a self-reported food frequency questionnaire.[18] In that analysis, the F-BA changed the eating behaviour for a number of food groups (eg, increasing fruit and vegetable consumption, whole grains and fish; decreasing the consumption of sugary snacks) but for not for others (eg, reducing meat, alcoholic beverages, high fat dairy and soft drinks). It should be noted that in the present study, the questionnaire used to assess the F-BA was tapping into beliefs about the successfulness of the strategies used while the food frequency questionnaire, although self-report, is effectively a count of actual food consumed over a given period. Future studies should include both subjective reports on interventions and more objective measures indicating behavioural change.

The overall evidence from the present analysis suggests that the F-BA would be acceptable to the public/patients in a way that the supplements would not. This is in keeping with the main trial results[17] which showed a placebo effect for the supplement trial arm for the secondary outcomes, indicating possible harmful effects, whereas the F-BA was effective in reducing symptoms of depression for those with higher severity at baseline and prevented incidence of depression at high levels of adherence to therapy. An important moderating factor on the effect of good F-BA adherence on depressive symptoms in the present study was having a past history of depression which suggests that future prevention approaches may be more effective when they adopt a more *selective* approach (eg, targeted at those with prior history or elevated risk of major depression), which is consistent with prior findings.[31] Taken as a whole, the present findings suggest that the F-BA has scope for further testing and development. It is however, too early to make recommendations for it to be rolled out as a public health prevention strategy in overweight adults with subsyndromal symptoms of depression. It is also important to carry out health economics evaluations for such an intervention. Most sessions of the F-BA are a 1:1 face-to-face intervention which involves trained psychologist time and may therefore be more costly to run than other interventions (eg, internet-based interventions).

The fact that participants reported preferring F-BA significantly more than daily supplement pills is also consistent with analogous research on patient therapy preference. For example, a meta-analysis has shown a

threefold patient preference for psychological over pharmacological treatments.[32] Similarly, there was a clear preference for individual over group F-BA therapy which is also consistent with previous research on mental health treatment preference in primary care.[33] It is important to take into consideration patient preference as it can predict good outcomes, as shown in a recent study that found that depressed patients who showed a stronger preference for a treatment (person-centred counselling vs low intensity cognitive behavioural therapy) fared better in that treatment compared with the alternative.[34]

It is also important to note that meta-analysis has shown smaller therapeutic effects for group versus individual modality in psychotherapy for depression[35] and this should be factored into the future development of prevention interventions. Individual F-BA sessions appear to be the best way to ensure such interventions are effective in real word settings.

We also found some evidence to suggest that F-BA session attendance was linearly related to the reduction in symptoms over time, only for those with a history of depression, partially replicating previous findings.[35] The a priori use of a threshold for good F-BA adherence (8/21 sessions) in our moderation analyses showed that for those with a history of depression, attending eight or more F-BA sessions predicted significant reductions in depressive symptoms. Future research using a similar intervention should therefore consider achieving eight sessions of intervention but continue to test where the best threshold lies. Future work should also continue to test the alternative hypothesis that change in depressive symptoms may affect adherence to intervention levels.

Similarly, in a meta-analysis on the treatment effect of mindfulness-based cognitive therapy in randomised clinical trials, the risk of relapse in depression was found to be larger in patients with higher initial depressive symptoms.[36] Although the sample in the MooDFOOD Trial was ostensibly a high-risk sample with elevated depressive symptoms and individuals being overweight or obese, the relatively low incidence of depression observed over 12 months (~10%)[17] suggests the inherent risk was not high. Future research should focus on higher-risk populations in order to improve the feasibility of interventions. For example, participants could be recruited with higher levels of depressive symptoms at baseline and a history of recurrent depression, which would likely confer a higher risk for subsequent depression.

Early work on the dose-response relationship has suggested that as few as 8 therapy sessions are sufficient for 50% of depressed patients to reach clinically significant change[37] although more recently there has been a general consensus that somewhere between 13–18 sessions are required.[38] Less is known on the optimal number of sessions for depression prevention, however meta-analytical reviews of intervention studies where the number of sessions ranged from 1 to 15, found a 21%–22% reduction in incidence of depression.[31 39] No dose-response relationship was found in either of the

above meta-analyses. An alternative explanation for the results is that those participants who attended no or few F-BA sessions differed in some essential way to those engaged more with the intervention. For example, initial depressive symptom levels may have been much higher for those individuals not engaging. However, we found no significant difference between the F-BA sessions attendance groups on any baseline characteristics including a history of depression, lending support to the interpretation that elements in the F-BA had caused the reduction of symptoms. Similarly, it is entirely possible and is intuitive, clinically, that early changes in depressive symptoms in the intervention may predict adherence. However, in the post hoc analysis we found no evidence for this association.

The results are also broadly consistent with current research on reducing depressive symptoms. For example, findings from a recent meta-analysis of 16 RCTs suggest that overall dietary interventions may have the potential to provide some benefit in reducing depressive symptoms.[40] In this review, the majority of studies (15/16) assessed non-clinical populations, as did the MooDFOOD Study. There was one notable exception, the Supporting the Modification of lifestyle In Lowered Emotional States (SMILES) Trial, which assessed dietary intervention in a clinical depression sample.[14] The intervention effect reported in this study was a very large difference in the reduction of depression symptoms for a dietary support group versus a social support group. A further recent trial has assessed the effect of a Mediterranean-style diet on depressive symptoms in depressed patients (the Healthy Eating for Life with a Mediterranean-style diet (HELFIMED) Trial).[15] This trial randomised patients to either a 'MedDiet' group receiving nutrition education, food hampers, cooking workshops and fish oil supplements for 6 months, or to fortnightly social groups for 3 months. The results showed statistically significant reductions in symptoms of depression for the MedDiet group. A further clinical trial on preventing the recurrence of clinical depression (the Nutritional Intervention With Mediterranean Diet in the Prevention of Recurrence of Depression (PREDI-DEP) Trial) is in progress and yet to report results. This intervention is comprised of a dietician-led Mediterranean-style diet supplemented with olive oil.[41]

A recent RCT using a brief dietary intervention (dietician-delivered intervention via a 13 min online video) reported a significant reduction in depressive symptoms for young people with elevated symptoms.[16] The results of this study are informative in that they suggest that nutritional interventions (brief ones) are feasible in young people with elevated depressive symptoms. Therefore, it is possible that a modified behavioural intervention such as the MooDFOOD F-BA may be efficacious in different populations, including younger people. Collectively, the results of these studies and the present analysis suggest that dietary interventions have the potential to reduce depressive symptoms, perhaps more so for

those with elevated symptoms, a current diagnosis of depression or for those with a history of depression.

## Limitations

The present study has a number of limitations. First, although the F-BA intervention was shown to be reasonably acceptable and feasible to deliver, the mediating mechanisms underlying reductions in depressive symptoms in this population are not yet known. Future research should aim to further elucidate the underlying mechanisms involved, and determine which individuals may benefit most from the intervention.[40] These should be teased out as it is possible that the responsible mechanisms are psychological, food-related or both and it is necessary to correctly identify the responsible mechanisms in order to ensure that the intervention is feasible, in the sense that it needs to be maximally effective. Second, it is not clear whether the F-BA approach to intervention is relevant for community samples with a full range of BMI values or only for overweight and obese individuals. This has important feasibility and policy implications and would influence the nature of prevention intervention potentially changing it from targeted or indicated to universal. A major limitation with the self-reported assessment of the F-BA was that this came only at the end of the trial. That is, we did not have intermediate evaluations. Although we were successful in reaching some participants at the end of the trial who had left the trial at an earlier point, it is entirely possible that a proportion of those who had dropped out of the trial and were unreachable at the end point held more negative views than those participants who responded. More broadly, the F-BA approach to depression prevention requires development and refinement through research to answer key questions on modality of delivery (eg, individual supported by web-based tools and mobile technology) and patient group (is there scope to roll out to different groups including chronic depression to prevent relapse, prevention in adolescents and young people?) before it can be considered for a larger-scale public health intervention. In conclusion, the present study showed that the F-BA depression prevention strategy was preferred by study participants over daily use of supplements, was well-received and largely considered acceptable. It has potential for good reach, was administered as intended, including a reasonable retention rate, and therefore has potential as a depression prevention intervention. Further work is needed before making clear recommendations for depression prevention in this and other groups.

**Author affiliations**
[1] Department of Psychology, University of Exeter, Exeter, UK
[2] Amsterdam UMC, Vrije Universiteit, Psychiatry, Amsterdam Public Health research institute, GGZ in Geest Specialized Mental Health Care, Amsterdam, The Netherlands
[3] Department of Health Sciences, Faculty of Science and Amsterdam Public Health research institute, Vrije Universiteit Amsterdam, Amsterdam, The Netherlands
[4] Institut Universitari d' Investigació en Ciències de la Salut (IUNICS/IDISBA), Rediapp, University of Balearic Islands, Palma de Mallorca, Spain
[5] Department of Psychiatry and Psychotherapy, University Leipzig, Medical Faculty, Leipzig, Germany
[6] Department of Psychiatry, Psychosomatics and Psychotherapy, Goethe-University Frankfurt, Frankfurt, Germany

**Collaborators** The MooDFOOD Prevention Trial Investigators: Bep Verkerk, post-BSc (data manager, GGZ inGeest Specialized Mental Health Care, Amsterdam, the Netherlands), Nadine Paans, MSc (field center therapist and research assistant, Amsterdam UMC, Vrije Universiteit, Amsterdam, the Netherlands), Carisha Thesing, MSc (field center therapist, Amsterdam UMC, Vrije Universiteit, Amsterdam, the Netherlands), Deborah Gibson-Smith, MSc (field center research assistant, Amsterdam UMC, Vrije Universiteit, Amsterdam, the Netherlands), Melany Horsfall, MSc (field center coordinator, GGZ inGeest Specialized Mental Health Care, Amsterdam, the Netherlands), Lena Weiss, MSc (field center research assistant, Amsterdam UMC, Vrije Universiteit, Amsterdam, the Netherlands), Amy Romijn, PhD (field postdoctoral research associate, University of Exeter, Exeter, United Kingdom), Hannah Bunce, MSc (field center associate research fellow), Owain Winfield, BSc (research therapist, University of Exeter, Exeter, United Kingdom), Harriet Bunker-Smith, MSc (field center associate research fellow, University of Exeter, Exeter, United Kingdom), Fern Durbridge, BSc (field center associate research fellow, University of Exeter, Exeter, United Kingdom), Caterina Versari Molinares, MSc (research intern, University of Exeter, Exeter, United Kingdom), Atikah Sapar, BSc (research intern, University of Exeter, Exeter, United Kingdom), Miquel Tortella, PhD (field center coinvestigator, UIB, Palma de Mallorca, Spain), Clara Homar Covas, MSc (field center research and therapist, UIB, Palma de Mallorca, Spain), M. M. Angeles Pérez-Ara, PhD (field center research assistant, UIB, Palma de Mallorca, Spain), Adoración Castro Gracia, MSc (field center research assistant, UIB, Palma de Mallorca, Spain), José Luis Reig, MSc (field center therapist, UIB, Palma de Mallorca, Spain), Jana Hoesel (field center study nurse, ULEI, Leipzig, Germany), Ezgi Dogan, MD (field center research fellow, ULEI, Leipzig, Germany), Sabrina Baldofski, PhD (field center therapist, ULEI, Leipzig, Germany), and Nicole Mauche, MSc (field center therapist).

**Contributors** MO, EW were the lead authors of the manuscript and MO analysed and interpreted the data with EW. MV and IAB obtained funding for the MooDFOOD project, designed the MooDFOOD prevention trial and together with MC coordinated the MooDFOOD project. BP, MB and EW contributed to the design of the MooDFOOD prevention trial. EW led the development and training of the MooDFOOD food-related behavioural change intervention. EK and UH coordinated the recruitment, interventions and follow-ups at the trial centre in Germany, University Leipzig. BP and MB coordinated the recruitment, interventions and follow-ups at the trial centre in the Netherlands, VU University Medical Center Amsterdam. EW and MO coordinated the recruitment, interventions and follow-ups at the trial centre in the UK, University of Exeter. MR and MG coordinated the recruitment, interventions and follow-ups at the trial centre in Spain, University of Balearic Islands. GvG set up the logistics for the trial's data collection. All authors contributed to the writing of the manuscript and approved the final version. Please, see www.moodfood-vu.eu for a complete list of the MooDFOOD prevention trial Investigators.

**Funding** This work was supported by the European Union FP7 MooDFOOD Project 'Multi-country collaborative project on the role of Diet, food related behaviour, and Obesity in the prevention of Depression' (grant agreement no. 613598). This work is supported in the UK by the National Institute for Health Research (NIHR), through the Primary Care Research Network and the NIHR Exeter Clinical Research Facility.

**Competing interests** MR reported receiving grants from the European Union and research funding from Janssen and Lundbeck outside the submitted work. BP reported receiving grants from Janssen Research and Boehringer Ingelheim. EW reported receiving royalties for a therapy manual in Behavioural Activation/Cognitive Behavioural Therapy from Guilford Press; and honorarium for running workshops in his rumination-focused cognitive behavioural therapy from different national cognitive behavioural therapy organisations worldwide. UH reported receiving a honorarium for scientific talks from Lundbeck, Janssen Pharmaceutica, and Servier, and a research grant from Medice outside the submitted work.

**Patient consent for publication** Not required.

**Ethics approval** The trial received approval from the local regional ethics committees.

**Provenance and peer review** Not commissioned; externally peer reviewed.

**Data availability statement** Data are available upon reasonable request. Data collected for this study will be made available after approval of an analysis plan by the MooDFOOD Trial publication committee. How to access data: Data can be accessed at https://www.moodfood-vu.eu/ or moodfood.po@vu.nl. Who can access

**ORCID iD**
Matthew Owens http://orcid.org/0000-0001-7407-6540

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
