## [Reviewer comments · BMJ Open]

ARTICLE DETAILS

TITLE (PROVISIONAL)	Acceptability and feasibility of two interventions in the MoodFOOD trial: a food-related depression prevention randomised controlled trial in overweight adults with subsyndromal symptoms of depression.
AUTHORS	Owens, Matthe; Watkins, Edward; Bot, M; Brouwer, Ingeborg; Roca, Miquel; Kohls, Elisabeth; Penninx, Brenda; van Grootheest, Gerard; Cabout, Mieke; Hegerl, Ulrich; Gili, Margalida; Visser, Marjolein

VERSION 1 – REVIEW

REVIEWER	Heather Francis Macquarie University
REVIEW RETURNED	29-Oct-2019

GENERAL COMMENTS	This paper reports on adherence to two nutritional strategies aimed at reducing symptoms of depression. This is particularly important work given the is substantial diet nihilism regarding a) whether the general public will adhere to diet recommendations and b) whether people with depression (given prominent symptoms of low motivation) will adhere to diet recommendations. The paper is well written and the findings are well described. I have two main concerns: 1) There appear to be some findings that have previously been reported on in the original publication of the trial outcomes. In particular, the diet adherence in the F-BA group of 71% is reported in the JAMA article, as well as the fact that individual sessions had better attendance than group sessions (despite different statistics being reported - median vs %). 2) In the discussion section, there could be more discussion of the implications of the specific findings, and more of a focus on the new rather than previously reported findings. Specific comments: Page 5 – it is noted that randomised controlled trials are required to test the causal direction of a relationship between diet and depression. It is worth noting that there are now 3 other RCTs to investigate a causal relationship: Jacka, F. N., O'Neil, A., Opie, R., Itsiopoulos, C., Cotton, S., Mohebbi, M., ... & Brazionis, L. (2017). A randomised controlled trial of dietary improvement for adults with major depression (the 'SMILES'trial). BMC medicine, 15(1), 23. Parletta, N., Zarnowiecki, D., Cho, J., Wilson, A., Bogomolova, S., Villani, A., ... & Segal, L. (2019). A Mediterranean-style dietary intervention supplemented with fish oil improves diet quality and mental health in people with depression: A randomized controlled trial (HELFIMED). Nutritional neuroscience, 22(7), 474-487.
--

	Francis, H. M., Stevenson, R. J., Chambers, J. R., Gupta, D., Newey, B., & Lim, C. K. (2019). A brief diet intervention can reduce symptoms of depression in young adults—A randomised controlled trial. PLoS one, 14(10), e0222768. Given the aim of the current study, it would be worth noting that diet adherence was reported in at least two of these studies. Though, more in depth analysis such as reported in the current manuscript has not yet been performed. Page 14 – under the heading F-BA adherence – Unless I am mistaken, the reported 71,3% adherence and greater attendance of individual sessions appears to have been previously reported in the JAMA article. There needs to be greater delineation of previously reported results. Page 18 – same comment regarding the first sentence of the discussion section. Page 19 – the structure of information could be improved here. Line 6 reports the finding that F-BA was more acceptable than supplements. This is brought up again on line 41. Line 20 – reports a history of depression moderated the effect of adherence on depressive symptoms. However this finding is not interpreted until line 46. Page 20 – “Individual F-BA sessions appear to be the best way to ensure such interventions are practically feasible”. Does the finding really suggest that group interventions are not practically feasible? Or is it that group sessions are not as appealing for some reason? Page 20 line 29 – “more effective when they are targeted rather than universal (aimed at the general population)”. I don’t know that this can be inferred from the finding – the RCT recruited people who were obese and with elevated depression symptoms on the PHQ-9. This already makes them more targeted than the general population.
--	--

REVIEWER	Grace Chan University of Connecticut School of Medicine USA
REVIEW RETURNED	20-Feb-2020

GENERAL COMMENTS	The authors investigated 1025 participants’ acceptability and level of compliance of their randomly assigned intervention arms and their effect on study outcome (depression symptoms) in the Multi-country cOllaborative project on the rOle of Diet, Food related behavior, and Obesity in the prevention of Depression (MooDFOOD) trial. After many exploratory analyses, they reported that individual food-related behavioral activation therapy (F-BA) sessions were better received than both group F-BA sessions and supplements (either multi-nutrient or placebo). However, F-BA adherence had no direct main effect on outcome, though its effect was moderated by participants’ past major depression episodes. Overall, the manuscript was well written and appropriate statistical analyses had been conducted. Nevertheless, there are many missing pieces in the main body of the manuscript. Please consider the following feedback:  1. Insufficient details of the MooDFOOD study  a. Please clearly state the number of subjects in each of the four arms as well as how many in each arm completed various post study questionnaires (F-BA overall, F-BA individual sessions, F-BA
--

	group sessions, supplements, etc.). b. Since the acceptability and adherence of different types of F-BA sessions were featured in the analyses, please provide the order/timing as well as topics covered in each of the 21 sessions (15 individual and 6 group). In particular, when were the eight strategies and ten targets discussed. c. Similarly, for the supplement intervention, please separately report the number and percentage of pills taken by subjects in each of the four arms. d. Assessment schedule for depression, diet, adherence over the 12-month treatment period is missing. In page 16, does T3 mean 3 months after baseline (T0)? If so, why was this time point selected here instead of at the end of treatment (T12)? 2. Many exploratory comparisons a. There does not appear that the reported results were corrected for multiple comparisons. Perhaps adjusted p-values should be used to determine statistical significance in this exploratory study. b. It is unclear how the reported comparisons were selected. For example, why were the following comparisons not discussed:  i. F-BA individual vs. nutrient supplement ii. F-BA individual vs. placebo supplement iii. F-BA group vs. nutrient supplement iv. F-BA group vs. placebo supplement 3. Minor issues. Only a couple of them are listed below:  a. References # 14 is the same as # 17 b. Reference # 15 is incomplete
--	---

VERSION 1 – AUTHOR RESPONSE

Reviewer 1:

Reviewer Name: Heather Francis

Institution and Country: Macquarie University

This paper reports on adherence to two nutritional strategies aimed at reducing symptoms of depression. This is particularly important work given the is substantial diet nihilism regarding a) whether the general public will adhere to diet recommendations and b) whether people with depression (given prominent symptoms of low motivation) will adhere to diet recommendations.

We thank the reviewer for their assessment of the research area and the present paper. We are also grateful for their helpful suggestions for improving the current manuscript.

The paper is well written and the findings are well described. I have two main concerns:

5. There appear to be some findings that have previously been reported on in the original publication of the trial outcomes. In particular, the diet adherence in the F-BA group of 71% is reported in the JAMA article, as well as the fact that individual sessions had better attendance than group sessions (despite different statistics being reported - median vs %).

The reviewer is correct that the overall adherence estimates are given in the JAMA paper, including the difference between individual and group F-BA sessions,

but we would like to point out that this only received a passing treatment in that paper. More detail on adherence is added here including analyses of a dose-response relationship, assessment of potential moderators and an analysis on the potential direction of causality between adherence and depressive symptoms. We strongly believe that the additional analyses are important and make a significant contribution to our understanding of psychological and nutritional intervention strategies in depression. Therefore, after careful consideration, we believe that the basic adherence estimates given in this paper should be retained in order to give context to the reader.

6. In the discussion section, there could be more discussion of the implications of the specific findings, and more of a focus on the new rather than previously reported findings.

We fully agree with the reviewer and have now rewritten the discussion to incorporate these points. In particular, we have removed some detail on the adherence estimates and added extensive discussion on how the novel results presented in this manuscript feed into the extant literature.

Specific comments:

7. Page 5 – it is noted that randomised controlled trials are required to test the causal direction of a relationship between diet and depression. It is worth noting that there are now 3 other RCTs to investigate a causal relationship:

Jacka, F. N., O’Neil, A., Opie, R., Itsiopoulos, C., Cotton, S., Mohebbi, M., ... & Brazionis, L. (2017). A randomised controlled trial of dietary improvement for adults with major depression (the ‘SMILES’ trial). *BMC medicine*, 15(1), 23.

Parletta, N., Zarnowiecki, D., Cho, J., Wilson, A., Bogomolova, S., Villani, A., ... & Segal, L. (2019). A Mediterranean-style dietary intervention supplemented with fish oil improves diet quality and mental health in people with depression: A randomized controlled trial (HELFIMED). *Nutritional neuroscience*, 22(7), 474-487.

Francis, H. M., Stevenson, R. J., Chambers, J. R., Gupta, D., Newey, B., & Lim, C. K. (2019). A brief diet intervention can reduce symptoms of depression in young adults—A randomised controlled trial. *PloS one*, 14(10), e0222768.

Given the aim of the current study, it would be worth noting that diet adherence was reported in at least two of these studies. Though, more in depth analysis such as reported in the current manuscript has not yet been performed.

We thank the reviewer for highlighting these more recent and important additions to the literature and we have now added these three studies to both the introduction and the discussion sections. We agree with the reviewer that while work has started on issues such as

adherence to nutrition-based interventions in depression, more in-depth analysis is needed and the current manuscript aims to add to this. We have added these points to the introduction immediately before the hypotheses.

8. Page 14 – under the heading F-BA adherence – Unless I am mistaken, the reported 71,3% adherence and greater attendance of individual sessions appears to have been previously reported in the JAMA article. There needs to be greater delineation of previously reported results.

The adherence estimates are included in the current paper only to add context to the novel in-depth analyses that build on those very basic estimates reported in the JAMA paper. We believe this minimal data on adherence from the JAMA paper should be retained so the reader can understand the context of the novel analyses presented in the current paper. We have also referred to the main trial paper for readers interested in further details on the trial more generally. The current paper focusses on a more in-depth exploration of the adherence and factors underpinning this. Thus, in our view, there is significant delineation between the two papers.

9. Page 18 – same comment regarding the first sentence of the discussion section.

In keeping with the principles noted above, we have now removed the opening sentence of the discussion.

Page 19 – the structure of information could be improved here.

10. Line 6 reports the finding that F-BA was more acceptable than supplements. This is brought up again on line 41.

We agree that the structure could be improved here. We have reworded the section starting on line 41 to better differentiate the point on patient preference from that of overall acceptability, made in line 6.

“The fact that participants reported preferring F-BA significantly more than daily supplement pills is also consistent with analogous research on patient therapy preference.” This and other changes can be seen in context in the marked copy document.

11. Line 20 – reports a history of depression moderated the effect of adherence on depressive symptoms. However this finding is not interpreted until line 46.

The moderation finding interpreted on line 46 has now been moved to line 20 to avoid confusion here and improve the clarity of expression in the paper: “An important moderating factor on the effect of good F-BA adherence on depressive symptoms in the present study was having a past history of depression which suggests that future prevention approaches may be more effective when they adopt a more selective approach (e.g. targeted at those with prior history or elevated risk of major depression), which is consistent with prior findings...”

12. Page 20 – “Individual F-BA sessions appear to be the best way to ensure such interventions are practically feasible”. Does the finding really suggest that group interventions are not practically feasible? Or is it that group sessions are not as appealing for some reason?

Yes, the issue here is more that the intervention is likely to be more effective with individual sessions. This has been added on page 20. It is likely that group sessions were not as appealing as individual ones, as you rightly point out. However, we do not have any data per se to speak to this issue. Anecdotally, some participants expressed the view that they preferred individual sessions because they could talk about personal issues in a way that they might not have been able to in group sessions, possibly due to feeling inhibited.

13. Page 20 line 29 – “more effective when they are targeted rather than universal (aimed at the general population)”. I don’t know that this can be inferred from the finding – the RCT recruited people who were obese and with elevated depression symptoms on the PHQ-9. This already makes them more targeted than the general population.

We agree with the reviewer on this point and have amended the text accordingly. It may be that such selective interventions have greater effectiveness when the risk factors are greater. That is, for example, if participants all have a history of clinical depression and have current symptoms at higher levels than was the case in the MoodFOOD trial.

Reviewer: 2

Reviewer Name: Grace Chan

Institution and Country:

University of Connecticut School of Medicine

USA

Please state any competing interests or state ‘None declared’: none

Please leave your comments for the authors below

The authors investigated 1025 participants’ acceptability and level of compliance of their randomly assigned intervention arms and their effect on study outcome (depression symptoms) in the Multi-

country collaborative project on the role of Diet, Food related behavior, and Obesity in the prevention of Depression (MooDFOOD) trial. After many exploratory analyses, they reported that individual food-related behavioral activation therapy (F-BA) sessions were better received than both group F-BA sessions and supplements (either multi-nutrient or placebo). However, F-BA adherence had no direct main effect on outcome, though its effect was moderated by participants' past major depression episodes.

Overall, the manuscript was well written and appropriate statistical analyses had been conducted. Nevertheless, there are many missing pieces in the main body of the manuscript. Please consider the following feedback:

We thank the reviewer for their considered review of the manuscript and insightful comments for improvement. We have carefully considered their feedback and address each of point in turn below.

14. Insufficient details of the MooDFOOD study

We have now added more details on the MooDFOOD study to the current manuscript including numbers of individuals in the four trial arms, inclusion of the assessment schedule, more details on the F-BA intervention and more detail on the rationale for the current paper, as well as more precision in the background to specific hypotheses. To be consistent with Reviewer 1 over the need to delineate between this and other publications coming out from our group, however, we believe that referring readers interested in finer details of the original trial to the JAMA MooDFOOD publication is a reasonable approach in this case.

15. Please clearly state the number of subjects in each of the four arms as well as how many in each arm completed various post study questionnaires (F-BA overall, F-BA individual sessions, F-BA group sessions, supplements, etc.).

We agree that adding this information on number of participants in each trial arm is important and we have now added this under Participants. More generally, missing data rates are reported in the main trial paper and we believe that additionally reporting them in the current manuscript would be repetitious. It is worth pointing out that participants in the MooDFOOD trial F-BA were scheduled to attend both individual and group therapy sessions, that is, they are not mutually exclusive. Missing data were accounted for in the current manuscript using a maximum likelihood estimator, which is a robust approach to analysing missing data.

16. Since the acceptability and adherence of different types of F-BA sessions were featured in the analyses, please provide the order/timing as well as topics covered in each of the 21 sessions (15 individual and 6 group). In particular, when were the eight strategies and ten targets discussed.

This is an important point raised by the reviewer and we have now outlined the timing and topics covered in the Method section. To answer the question directly here, the strategies were covered in the initial sessions but reinforced throughout the intervention.

“F-BA was provided by trained psychologists familiar with BA (15 individual sessions, 6 group sessions, for a one-year period) and a dietician was available to all study sites to provide advice. The intervention was delivered over a maximum of 21 sessions with up to 15 half-hour individual sessions, provided in single or double (one-hour) meetings. Individual sessions were initially given weekly (sessions 1-8) and then held every two weeks (sessions 9-15). These were followed by up to six group sessions (maximum of 10 participants per group and lasting approximately one hour) occurring monthly initially (sessions 16-18) and finally once every two months (sessions 19-21). A range of potential strategies were employed including food-related strategies and food group targets to aim for. These were introduced and explained to participants in the course of the first eight sessions and subsequently revisited and reviewed over the remaining sessions. F-BA followed principles of introducing participants to key information and then reinforcing and consolidating concepts over time.”

17. Similarly, for the supplement intervention, please separately report the number and percentage of pills taken by subjects in each of the four arms.

This information adds further to the basic adherence that we included in this paper for the readers' context. This information is reported in the main trial paper and so to avoid overlap with that, we believe it is better not to report this level of detail in the present paper.

18. Assessment schedule for depression, diet, adherence over the 12-month treatment period is missing. In page 16, does T3 mean 3 months after baseline (T0)? If so, why was this time point selected here instead of at the end of treatment (T12)?

We thank the reviewer for pointing out that an assessment schedule was not included in the original manuscript. We agree that it should have been and have now included the measurement schedule under the Measures section. “Measures were collected at baseline (T0), three months (T3), six months (T6) and at 12 months (T12; end of trial).” Yes, we can confirm that you are right to say that T3 refers to the timepoint three months after baseline (T0) and this is now made explicit in the text. The reason for analysing change from T0 to T3 in this case was to allow us to test an alternative hypothesis that change in depressive symptoms would lead to poor adherence to intervention over the trial. This was not clear in the original manuscript but we have now added this to the hypothesis section. “We also specified a model that allowed us to test an alternative hypothesis that change in depressive symptoms would lead to poor adherence to intervention over the trial. In this model, change in depressive symptoms was specified from T0 to the earliest possible timepoint (T3) to maximise the temporal precedence of depressive symptoms over adherence.”

REVIEWER	Grace Chan University of Connecticut School of Medicine USA
REVIEW RETURNED	19-Apr-2020

GENERAL COMMENTS	This version is much better than the previous version. There are still a few needed improvements. See suggestions below:  1. Have the authors considered two separate F-BA intervention adherence measures: one among the 15 individual sessions and another one among the 6 group sessions? Moreover, when reporting “Dose-response effect of F-BA therapy”, please include separated and combined descriptive statistics on the number of individual and group sessions. Considering the differences in how participants felt about these two session formats, these more fine-grained measures should provide much needed insight into the design of subsequent session formats. The authors might need to use Poisson or Negative Binomial regressions to model change in depressive symptoms on the number of individual and/or group F-BA sessions attended instead of normal linear regression up to the third degree. Also, please clearly state if the number of group F-BA sessions (or some binary versions) was controlled for when examining the dose-response effect of the number of individual F-BA sessions on change in depressive symptoms and vice versa. 2. Thank you for clarifying the analysis using “change in depressive symptoms” from T0 to T3. However, instead of evaluating this as a potential predictor for “adherence to intervention over the trial”, which COMBINED the same period T0 to T3 and subsequent time points (T3 to T12), separating this analysis into two: one for evaluating the association between “change in depressive symptoms” from T0 to T3 and adherence over the same period (i.e., from T0 to T3 only); and another for evaluating the predictive power of “change in depressive symptoms” from T0 to T3 on adherence afterward (i.e., from T3 to T12), would provide more clinically relevant findings. Also please explain the current hypothesis: “to test an alternative hypothesis that the change in depressive symptoms would lead to poor adherence to intervention over the trial.” Did “change” refer to improve or worsen symptoms? Were the authors hypothesizing that if there is NO CHANGE in depressive symptoms by T3, then participants were expected to attend more F-BA individual and/or group sessions? Why? 3. Was the statistically significant change in PHQ-9 (total score range = 0 – 27) of -0.08 (SE = 0.03) clinically meaningful? A few more minor issues or typos:  1. P. 15 “Multi-nutrient supplement and placebo pills” subsection: The degrees of freedom was missing for the χ^2 test with test statistics = 146.40 and p-value < 0.001. Most of the other test results in text were also in Table 2 with degrees of freedom, but not this one. Please either add the degrees of freedom in text or add the corresponding comparison to Table 2. 2. P. 15 “F-BA adherence” subsection: Should the word “group” in the phrase “attended 1 or more group sessions” be replaced by “individual”? 3. Table 1 third last line: Remove the extra “green label?” in column 1. 4. Table 2: Add the sample sizes for “pills” and “Placebo”.
--

VERSION 2 – AUTHOR RESPONSE

Reviewer: 2

Reviewer Name: Grace Chan

Institution and Country:

University of Connecticut School of Medicine

USA

Please state any competing interests or state 'None declared': none

Grace Chan

Institution and Country

University of Connecticut School of Medicine

USA

Please state any competing interests or state 'None declared':

None

Please leave your comments for the authors below

This version is much better than the previous version. There are still a few needed improvements. See suggestions below:

1. Have the authors considered two separate F-BA intervention adherence measures: one among the 15 individual sessions and another one among the 6 group sessions? Moreover, when reporting "Dose-response effect of F-BA therapy", please include separated and combined descriptive statistics on the number of individual and group sessions. Considering the differences in how participants felt about these two session formats, these more fine-grained measures should provide much needed insight into the design of subsequent session formats. The authors might need to use Poisson or Negative Binomial regressions to model change in depressive symptoms on the number of individual and/or group F-BA sessions attended instead of normal linear regression up to the third degree. Also, please clearly state if the number of group F-BA sessions (or some binary versions) was controlled for when examining the dose-response effect of the number of individual F-BA sessions on change in depressive symptoms and vice versa.

RESPONSE: The F-BA package was designed to start with individual sessions, then leading on to group sessions, with the former focused on individualised review of food and mood habits and planning of new approaches, and the latter as consolidation and maintenance of what was learnt. With respect to the issues of adherence in individual and group sessions, it is not possible calculate separate formal adherence estimates for individual and group sessions, respectively. This is because the F-BA protocol was designed in such a way that individual sessions were always followed by group sessions. That is, users did not have a choice between individual or group sessions; they were not run in parallel. Therefore time and modality (individual sessions vs group sessions) are confounded. This was stated on p.9 of the manuscript and we have now added the following clarifying text:

VERSION 3 – REVIEW

REVIEWER	Grace Chan University of Connecticut School of Medicine
REVIEW RETURNED	05-Jun-2020
GENERAL COMMENTS	Thank you for addressing all the comments.